# Sodium Solid Electrolytes: Na_x_AlO_y_ Bilayer-System Based on Macroporous Bulk Material and Dense Thin-Film

**DOI:** 10.3390/ma14040854

**Published:** 2021-02-10

**Authors:** Antonia Hoppe, Cornelius Dirksen, Karl Skadell, Michael Stelter, Matthias Schulz, Simon Carstens, Dirk Enke, Sharon Koppka

**Affiliations:** 1Institute of Chemical Technology, Universität Leipzig, 04109 Leipzig, Germany; antonia.hoppe@uni-leipzig.de (A.H.); simon.carstens@uni-leipzig.de (S.C.); dirk.enke@uni-leipzig.de (D.E.); 2Fraunhofer Institute for Ceramic Technologies and Systems IKTS, Michael-Faraday-Str. 1, 07629 Hermsdorf, Germany; cornelius.dirksen@ikts.fraunhofer.de (C.D.); karl.skadell@ikts.fraunhofer.de (K.S.); michael.stelter@ikts.fraunhofer.de (M.S.); matthias.schulz@ikts.fraunhofer.de (M.S.)

**Keywords:** sol-gel, Na-β-alumina, bilayer electrolyte, sodium-ion batteries

## Abstract

A new preparation concept of a partially porous solid-state bilayer electrolyte (BE) for high-temperature sodium-ion batteries has been developed. The porous layer provides mechanical strength and is infiltrated with liquid and highly conductive NaAlCl_4_ salt, while the dense layer prevents short circuits. Both layers consist, at least partially, of Na-β-alumina. The BEs are synthesized by a three-step procedure, including a sol-gel synthesis, the preparation of porous, calcined bulk material, and spin coating to deposit a dense layer. A detailed study is carried out to investigate the effect of polyethylene oxide (PEO) concentration on pore size and crystallization of the bulk material. The microstructure and crystallographic composition are verified for all steps via mercury intrusion, X-ray diffraction, and scanning electron microscopy. The porous bulk material exhibits an unprecedented open porosity for a Na_x_AlO_y_ bilayer-system of ≤57% with a pore size of ≈200–300 nm and pore volume of ≤0.3 cm^3^∙g^−1^. It contains high shares of crystalline α-Al_2_O_3_ and Na-β-alumina. The BEs are characterized by impedance spectroscopy, which proved an increase of ionic conductivity with increasing porosity and increasing Na-β-alumina phase content in the bulk material. Ion conductivity of up to 0.10 S∙cm^−1^ at 300 °C is achieved.

## 1. Introduction

Electrochemical energy storage is a key feature in transforming the global economy towards sustainability and carbon neutrality. The main alternative to Li-ion based batteries (LIBs), which are facing problems in terms of ethical issues and expensive lithium and cobalt production, are Na-ion based batteries (SIB). Sodium is cheap, environmentally safe and a suitable replacement for lithium [1,2].

The only industrially established sodium-based batteries are high-temperature Na/NiCl_2_- and Na/S-batteries. In both technologies, a non-porous ceramic Na-β-alumina tube is used as an electrolyte [3]. Polycrystalline Na-β-alumina electrolytes contain the two crystalline phases Na-β’-alumina (hexagonal, P63/mmc) and Na-β”-alumina (rhombohedral, R3m), which differ in stacking sequence and chemical composition [3]. The ionic conductivity of both crystal structures is caused by a conduction layer, where the mobile Na-ions are located on the c axis, and can, hence, move along this direction. [4,5] Thereby, an ionic conductivity of >0.2 S∙cm^−1^ at 300 °C can be reached [3,6]. Other recent studies undermine the general research interest in sodium-ion batteries [7,8].

The work presented in this article pursues the idea of a partially porous solid-state bilayer electrolyte for Na-ion based batteries. The general working principle is illustrated in Figure 1.

The left side of Figure 1 shows the working principle of a state-of-the-art Na/NiCl_2_-cell. Sodium serves as an anode and is labeled green. The active material (labeled black) is infiltrated with a highly conductive liquid electrolyte (NaAlCl_4_, labeled blue). This ensures a conductive connection between the particles. The separation of anode and cathode is realized by a dense Na-ion-conductive Na-β-alumina layer (labeled dark grey). The thickness of commercial Na-β-alumina electrolytes amounts from 1.5 to 2 mm. This guarantees a sufficient fracture strength to withstand the thermally and mechanically induced stress at operating temperatures of 250 °C up to 330 °C [9]. The state-of-the-art industrial manufacturing route of dense Na-β-alumina electrolytes represents a solid-state synthesis as powder followed by isostatic pressing of green bodies and subsequent sintering [10]. Lab-scale syntheses of Na-β-alumina powder are established for several methods: Solid-state reaction [11,12], infiltration method [13], and citrate complexation (also called sol-gel synthesis) [14,15,16,17]. Dense Na-β-alumina electrolytes are obtained after fractionation, uniaxial pressing, and sintering [1,6,12,16,17,18,19].

The aim of this work is to develop a novel, partially porous solid-state bilayer electrolyte (cf. Figure 1, right). The thicker layer (labeled light grey) is highly porous and infiltrated with liquid NaAlCl_4_. To prevent short circuits, the porous layer is coated with a thin, dense layer of Na-β-alumina. Due to this setup, the transportation of Na-ions through the electrolyte is divided into two parts. The transport across the porous area, ending at the dense Na-β-alumina layer, is realized by the liquid NaAlCl_4_ salt and partly by the porous structure itself, while the transport through the dense thin layer is accomplished by solid-state Na-β-alumina. Since the conductivity of NaAlCl_4_ at 300 °C is about three times higher than the conductivity of Na-β-alumina [20], the overall resistance of the bilayer electrolyte can potentially be significantly lowered in comparison to a state-of-the-art dense Na-β-alumina electrolyte with the same thickness.

To the best of our knowledge, only one publication shares this idea. Jung et al. [21] prepared a thin solid electrolyte consisting of a porous layer of 250 µm and a dense layer of 125–250 µm. Firstly, a stack of two α-Al_2_O_3_/ZrO_2_ layers was prepared by tape casting. In one layer, graphite powder is implied as pore forming agent. Both layers were laminated together and calcined at 1450 °C, while embedded in NaAlO_2_/Na-β”-alumina powder. Due to Na-diffusion from the bed into the tape casted α-Al_2_O_3_/ZrO_2_ foil, a reaction to a Na-β-alumina/ZrO_2_ took place. This bilayer electrolyte was placed in a cell similar to the principle described in Figure 1. Contrary to our approach Jung et al. placed the porous side at the anode and filled the pores with liquid sodium instead of NaAlCl_4_. To overcome the high surface tension of liquid sodium, which hinders the infiltration of the pores, lead acetate was used to coat the porous side [21]. They showed that using a bilayer electrolyte could improve the energy efficiency of a Na/NiCl_2_-cell up to 8%. Only a few other publications are available on porous Na-β-alumina. For instance, Fukui et al. [22] prepared a porous Na-β-alumina disc with a diameter of 15 mm via a solid-state reactive sintering process. The disc has a modal pore size of 630 nm with a pore volume of 0.20 cm^3^∙g^−1^, and total porosity of up to 58%. Sakka et al. [23] infiltrated porous α-Al_2_O_3_ with a sodic solution. After sintering at 1600 and 1700 °C, porous Na-β-alumina with an inhomogeneous pore structure was formed. While no reliable data on the costs of the aforementioned methods in comparison with our sol-gel synthesis is available yet, slurry casting [21] is certainly a less time-consuming approach. However, the sol-gel synthesis allows for a controlled interconnected pore structure with the potential of monolith production [24].

Our work pursues the novel idea of porous Na-β-alumina generation via the epoxide-mediated sol-gel synthesis, which is known from the synthesis of crystalline alumina with an interconnected pore structure [25]. To obtain a porous monolith instead of a non-porous powder, the pH value of an alcoholic solution containing an Al^3+^ salt must be increased slowly [24]. Propylene oxide (PO) is used to induce the hydrolysis and condensation of solvated Al^3+^-ions by an exothermic ring opening reaction. This enables the production of gels with an interconnected pore structure. Tokudome et al. [24] introduced polyethylene oxide (PEO) into the sol-gel synthesis of alumina. PEO induces phase separation, which results in materials with adjustable macropore sizes from 400 nm to 1.8 µm.

In this report, a three-step procedure is described to produce a partially porous solid-state bilayer electrolyte (BE). These steps consist of a sol-gel synthesis, the preparation of a disc from the porous, calcined bulk material, and spin coating to deposit a dense layer onto the thus obtained disc. This approach is fundamentally different from the work previously reported by Jung and co-workers, both regarding the preparation method and the envisaged results in terms of quantifiable porosity and Na content. The effect of methanol, sodium cations, and the concentration of PEO on the pore and crystal formation of the bulk material is investigated. Furthermore, the causal connections of crystal phase composition, pore structure, and the sintering process are discussed. Ultimately, the BEs are characterized regarding their ionic conductivity.

## 2. Materials and Methods

### 2.1. Calcined Xerogel Disc via Epoxide-Mediated Sol-Gel Synthesis, Powder Preparation, and Isostatic Pressing

The macroporous sodium-containing alumina bulk material was synthesized via an epoxide-mediated sol-gel route as described by Carstens and Enke [26]. The synthesis procedure was slightly modified regarding the replacement of ethanol by methanol to enable the integration of sodium-ions in the system.

7.80 g AlCl_3_·6H_2_O (Alfa Aesar, 99%) and 0.00, 0.01 or 0.10 g polyethylene oxide with a molecular weight (M.W.) of 900,000 (Acros Organics) were dissolved in a mixture of 5.77 g deionized water and 7.90 g methanol (VWR Chemicals, 100.0%) at room temperature. The solution was placed in an ice bath and cooled down to 5 °C within 1 h. 1.414 g of 4 M NaOH (VWR chemicals) were added under constant stirring (500 rpm). Immediately, 7.00 mL propylene oxide (Acros Organics, 99.5%) was added. The ice bath was removed after three minutes. The reaction solution was stirred for another seven minutes at room temperature. The reaction vessel was then placed in a water bath at 40 °C to induce gelation, followed by aging for 24 h at 40 °C. After an ethanol exchange for five days and drying at 50 °C for seven days, the obtained xerogels were calcined at 1200 °C for 1 h with a heating rate of 10 K∙min^−1^ (IH 60/14, Nabertherm, Germany). For disc preparation (diameter: 20 mm, high: 3 mm), the calcinated xerogels were ground to a particle size <350 µm and pressed into a disc form.

### 2.2. Dense Thin Layer via Spin Coating

The slurry was produced by ball milling (400 rpm; 20 min; 30 g of 3 mm-ZrO_2_-milling balls) of 9.5 g Na-β-alumina [6], 0.5 g TiO_2_ (Alfa Aesar, Haverhill, MA, USA 99.7%), 1 g Y-stabilized ZrO_2_ (Tosoh Corporation, Tokyo, Japan), 14.75 mL deionized water and 0.45 g organic binder. The calcined xerogel discs were spin coated (POLOS Spin150i, Putten, The Netherlands) according to the spinning regime in Appendix A.

After coating, the samples were dried overnight under ambient conditions. The dried samples were placed in a dense MgO-crucible with the coated layer facing upwards. A thin layer of Na-β-alumina powder was added to the bottom of the crucibles to avoid shrinkage deformation of the samples during heat treatment. Sintering was performed at 1400 °C for 2 h.

To guarantee a sufficiently dense layer, the procedure was repeated three times. It should be mentioned that the powder layer was only applied for the first sintering process.

### 2.3. Synthesis of NaAlCl_4_

NaAlCl_4_ was synthesized by stirring AlCl_3_ (BASF SE, Ludwigshafen, Germany) and NaCl (Akzo Nobel, Amsterdam, The Netherlands) in a molar ratio of 1:1.05 under N_2_ atmosphere at 180 °C for 1 h. The XRD pattern of the synthesized NaAlCl_4_ is shown in Appendix A.

### 2.4. Characterization Methods

The gelation process was controlled by temperature and pH-measurement (Mettler Toledo Seven Easy pH, Sensor InLab^®^ Routine Pro, Gießen, Germany).

The thermal properties of non-calcinated, powdered xerogels (<350 µm) were characterized by thermogravimetric analysis (TG/DSC, STA 409, Netsch, Selb, Germany). Thirty milligrams powder was placed in a corundum crucible. Corundum powder was used as a reference. The measurement was carried out from room temperature to 1300 °C with a heating rate of 10 K∙min^−1^ under atmospheric air.

The chemical composition was determined by inductively coupled plasma-optical emission spectroscopy (Optima 800, Perkin Elmer, Waltham, MA, USA). 50 mg of the powdered sample were mixed with 2 mL H_2_SO_4_ (93–98%, NORMATOM, VWR Chemicals, Radnor, PA, USA) and 0.75 mL H_3_PO_4_ (85%, suprapur, Merck, Darmstadt, Germany), and solubilized in a microwave oven with an 8 XF100 rotor at 1300 W (Multiwave Pro, Anton Paar, Graz, Austria). This solution was analyzed.

The carbon content was determined by CHN elemental analysis with a Vario EL III (Elementar Analysesysteme GmbH, Langenselbold, Germany). 30 mg of the powdered sample were heated up to 1800 °C and completely oxidized to CO_2_ on a WO_3_-catalyst.

Powder X-ray diffraction (XRD) analysis was carried out with a D8 DISCOVER (Bruker Corporation, Billerica, MA, USA). CuKα radiation (40 kV, 40 mA) and a VANTEC-500 2D detector (Bruker Corporation, Billerica, MA, USA) were used. A 2θ-correction was carried out about 0.1° regarding the reference measurement of corundum. The evaluation was carried out by Match! (version 3.3.0, Crystalimpact, Bonn, Germany). The quantification of the relative number of crystal phases was carried out by Rietveld analysis by applying Topaz software (Bruker, Billerica, MA, USA). The following data were used: ICSD-99830 (γ-Al_2_O_3_), ICSD-66560 (θ-Al_2_O_3_), ICSD-30024 (α-Al_2_O_3_), ICSD-201178 (Na-β-alumina).

The xerogel microstructure was characterized by scanning electron microscopy (LEO GEMINI 1530, Zeiss, Ulm, Germany). The samples were vapor coated with gold and measured with a high electron tension of 20.00 kV at a working distance of 14.5 mm.

The microstructure of the bilayer electrolytes was characterized after embedment in epoxy and polishing by scanning electron microscopy (Ultra 55 plus, Zeiss, Germany) at a voltage of 10 kV. Furthermore, energy-dispersive X-ray spectroscopy (Trident XM14, EDAX, Weiterstadt, Germany) was employed. Both methods were applied before the liquid salt infiltration.

The pore size distribution, porosity, and pore volume were measured via mercury intrusion (PASCAL 440 by ThermoScientific/POROTEC, Waltham, MA, USA). The samples were evacuated to 0.2 mbar prior to filling with mercury. Intrusion measurements were carried out up to 400 Mpa at room temperature with an assumed contact angle of mercury of 140° and its surface tension set to 0.48 N∙m^−1^. The density of each sample was determined by helium pycnometry (Multi Nr. 160 494, ThermoScientific/Pycnomatic ATC, Waltham, MA, USA) to calculate the porosity from the mercury intrusion data.

The sintered bilayer electrolytes were attached to an Al_2_O_3_-cylinder to measure the ionic conductivity. To check the quality of the dense layer and the glass joint, the electrolyte assemblies were tested using a leak detector (Pfeiffer Vacuum Smart Test, Asslar, Germany) and considered as gas tight, if the leakage rate was below 1 × 10^−7^ mbar∙L∙s^−1^. Na (VWR 99.8%, Radnor, PA, USA) was applied on the dense side, while the porous side was infiltrated with NaAlCl_4_ at 300 °C. Both electrolyte sides were contacted with a Cu sheet. The flat test cell was sealed with Teflon^TM^ (NaAlCl_4_-side) and carbon (Na-side). The ohmic resistance was measured by impedance spectroscopy (Biologic SP-240, Claix, France) from 1 MHz to 10 Hz and a sinus amplitude of 10 mV at temperatures from 165 to 300 °C. The cell assembly and measurements took place under N_2_ atmosphere. The impedance data were recorded and processed with the software “EC-Labs 11.3”. Equation (1) was used to calculate the specific ohmic conductivity (σ) of the electrolyte:(1)$$\sigma \;=\;\frac{t}{A\cdot R}$$
“*A*” represents the average cross-sectional area, “*R*” the measured ohmic resistance, and “*t*” the thickness of the electrolyte.

## 3. Results

### 3.1. Synthesis and Properties of Xerogels

#### 3.1.1. Characterization of the Sol-Gel Synthesis

The sol-gel process strongly depends on pH and temperature. Figure 2 shows the pH and temperature progress of a sample without PEO.

The AlCl_3_ solution has an initial temperature of 5 °C and a pH value of ≈0.4 (t = “−1” min). After sodium hydroxide addition, these values rise immediately to 1.7 and 8 °C, respectively (t = 0 min). After PO addition, the temperature and pH slowly continue to increase. Gelation is observed after 20 min. After the aging process (40 °C, 24 h) an intact lyogel is obtained, which cracks during drying (Figure 3, right). Temperature, pH progression, gelation time, and crack formation are similar for all samples.

Temperature and pH progression are similar to the study of Carstens and Enke [26]. In comparison, the addition of ≈5 wt.-% sodium hydroxide increases the pH value. PO raises the pH continuously, but with a smaller increase. Due to slightly less amount of water (−4 wt.-%), hydrolysis and exothermic ring opening reactions are slowed down. The temperature increases continuously. To ensure a controlled reaction, the velocity of hydrolysis and condensation is reduced by cooling. The latter results in similar temperature values. Gelation is observed at the same time. No influence of PEO (<0.35 wt.-%) on the sol-gel synthesis is identified via temperature and pH progression.

#### 3.1.2. Thermal Properties of the Xerogels

The thermal properties of the obtained xerogels are characterized by TG/DSC up to 1300 °C (Figure 4). The uncertainty of the TG measurement is ≤0.5 wt.-%.

For all xerogels, a loss of mass of ≈66 wt.-% is detected. The main mass loss takes place between 25 °C and 600 °C (Figure 4a), with one change in slope at around 200 °C. The mass loss of the samples with PEO is delayed in the temperature range above 200 °C. The theoretical mass loss of PEO (≤0.35 wt.-%) is within the defined uncertainty of measurement.

An endothermic process takes place between 25 °C and 170 °C (Figure 4b). A plateau-like transition (170–280 °C) is followed by exothermic processes (280–600 °C) with three local minima at 290, 390 and 480 °C. Between 1200 °C and 1250 °C, another exothermic process occurs. The addition of PEO reduces the intensity of the three exothermic processes at 230–600 °C. Furthermore, 0.10 g PEO shifts the exothermic process at 1200 °C to lower temperatures (Appendix A).

The first endothermic process (25–170 °C) can be attributed to the mass loss of adsorbed water. The exothermic processes (170–600 °C) can be attributed to the combustion of PEO and the phase transformation of alumina species. At 300 °C Hill et al. [27] reported the conversion of Al(OH)_3_ to γ-AlO(OH) (boehmite). The further conversion to γ-Al_2_O_3_ is usually observed between 300 °C and 500 °C [27,28]. Both phase transformations (280–600 °C) are accompanied by the release of chemically bound hydroxyl groups. The continuous mass loss above 600 °C is due to the release of residual hydroxyl groups. Between 600 °C and 1200 °C, no thermodynamic process is observed. In contrast, the literature reports a phase conversion to θ-Al_2_O_3_ between 850 °C and 1100 °C. At 1200 °C phase conversion of water-free Al_2_O_3_ phases, e.g., to α-Al_2_O_3_ or Na-β-alumina formation occurs [27,28].

PEO may reduce the number of crystal phases formed between 230 °C and 600 °C and could influence the phase formation process at 1200 °C. Nevertheless, further investigations are required to identify the effect of PEO on phase formation (cf. Section 3.2).

### 3.2. Characterization of Calcined Xerogels

#### 3.2.1. Characterization of Chemical and Phase Composition

All three lyogels consist of 2.4 wt.-% Na_2_O and 87.6 wt.-% Al_2_O_3_. The Na_2_O content is reduced to 1.1 wt.-% after the ethanol exchange and remains constant during further heat treatment. The Na_2_O content is independent of the initial PEO content. PEO is virtually completely combusted during calcination, and the final carbon content amounts to only 0.22 wt.-% (±0.10%).

The initial amount of sodium of 13.4 mol-% (or 8.63 wt.-%) is based on the ideal structural formula of Na-β”-alumina (Na_2_O·5.33Al_2_O_3_) and is limited by the solubility in the methanol-water solvent mixture [15,22]. The largest sodium loss (−6.23 wt.-%) occurs during sol-gel synthesis. The lyogel consists of water-rich alumina species and solvent. Sodium is predominantly present in the solvent phase and is only partially incorporated into the Al_2_O_3_ network. Although sodium salts are very poorly soluble in ethanol, sodium is physically washed out (−1.3 wt.-%) by the solvent exchange. After the calcination, the phase structures of calcined xerogels were characterized via XRD (cf. Figure 5).

All XRD patterns exhibit a broad diffuse background and broadened reflexes. Major reflexes of γ-Al_2_O_3_ and θ-Al_2_O_3_ can be identified_._ Qualitatively, the relative intensities remain similar among all three samples. The sample synthesized without PEO shows sharp reflexes of α-Al_2_O_3_ at 25.5°, 43.2°, and 57.4° 2θ. The first one disappears with the addition of 0.01 g PEO. With an initial amount of 0.10 g PEO, the latter two are nearly completely erased. A sodium-containing phase could not be identified.

The samples form sodium-free crystal structures of alumina although 1.1 wt.-% Na_2_O is contained. This can be explained by the incomplete phase conversion. In agreement with the TG/DSC measurement, the conversion to γ-Al_2_O_3_ is confirmed. Although no θ-Al_2_O_3_ phase conversion was identified by thermal analyses between 850 and 1000 °C, all samples contain θ-Al_2_O_3_ according to XRD. This phase conversion may have taken place prematurely below 600 °C, or shifted upwards to the temperature range of 1200–1250 °C. It is reported that θ- and α-Al_2_O_3_ can be formed simultaneously at 1100 °C [28]. Although α-Al_2_O_3_ is only present in traces in the sample, the exothermic phase transition at 1200 °C is clearly visible in the DTA curves in Figure 4. Formation of θ-Al_2_O_3_, thus, probably occurs at 1200 °C, along with α-Al_2_O_3_.

The α-Al_2_O_3_ content is reduced with increasing initial PEO amount in the reaction solution. Guzmán-Castillo et al. [29] reported that the crystallization and the conversion to transitional alumina (e.g., γ-, δ-, and θ-Al_2_O_3_) are influenced by the crystallite size of boehmite. With increasing grain size, the removal of hydroxyl groups becomes more difficult and results in a poorer conversion to transitional alumina, subsequently also to α-Al_2_O_3_. The XRD measurements illustrate that initial PEO inhibits the α-Al_2_O_3_ conversion. PEO is burned out during thermal treatment around 200 °C before Al_2_O_3_ phase conversion occurs. Thus, PEO could be responsible for increased boehmite crystallite sizes, resulting in a lesser conversion towards transition alumina and finally α-Al_2_O_3_. The latter phase is only visible via XRD studies. This assumption is supported by the PEO-induced intensity reduction of exothermic processes at 230–600 °C in the DTA curves in Figure 4.

#### 3.2.2. Pore Structure and Porosity Studies of Xerogels

Appendix A summarizes the texture data from the mercury intrusion of calcined xerogels, and Figure 6 shows a representative example of the pore size distribution.

The mercury intrusion data do not differ between the samples (cf. Appendix A). All samples exhibit a monomodal, open-pore system with a modal pore size of 21 nm, a pore volume of 0.4 cm^3^∙g^−1^ and a porosity of around 60%. A further assessment of the texture properties can be made by SEM images in comparison to mercury measurement. The SEM images in Figure 7 reveal the presence of macropores, which are not detected by mercury intrusion. The samples are distinguished based on the amount of PEO, even though the polymer was burned out by the thermal treatment.

Without PEO, the sample possesses a homogeneous structure, which is interrupted by isolated areas with interconnected pores. These areas are sporadically and statistically placed. The microstructure remains similar after the addition of 0.01 g PEO. The sample initially containing 0.10 g PEO shows a coarser material structure with homogeneously distributed semi-spherical cavities of about 1–2 µm in diameter and no additional isolated areas with interconnected pores.

Through the sol-gel and drying process, a pore system develops within the alumina network [30]. Due to calcination the system partially crystallizes. The homogeneous structure can be attributed to the part containing a homogenous mesoporous alumina network formed by γ- and θ-Al_2_O_3_ phases. Phase conversion from boehmite via transition alumina to α-Al_2_O_3_ induces sintering, resulting in increased pore sizes [25,29,31]. Therefore, the isolated areas can, hence, be plausibly attributed to α-Al_2_O_3_. The isolated semi-spherical cavities resulting from the addition of 0.10 g PEO to the synthesis mixture cannot be detected by mercury intrusion. Most of the cavities are only accessible in the complex pore structure via the smaller mesopores with 21–22 nm diameter. That means, the cavities are filled with mercury at a pressure that corresponds to the filling of the smaller mesopores (network effect of mercury intrusion). Therefore, the cavities are not displayed in the pore size distribution.

An intriguing question is why PEO appears to form those semi-spherical cavities within the gel network, rather than interconnected cavities [25,26] (cf. Figure 7c). For aluminum salt-based gels, an amount of ≥0.1 wt.-% PEO (M.W. 900,000) usually results in an additional interconnected macropore system around 1000 nm [20,21]. PEO induces phase separation at an early state of reaction, forming alumina (AH), and a PEO rich phase. Only week interactions might exist between AH and PEO. The solvent coexists in both phases [32].

PEO-induced phase morphology is determined by the mechanism (binodal, spinodal), the time difference (Δt) of the onset of phase separation, and sol-gel transition (SGT). The morphology is coarsened with increasing Δt and frozen at SGT. The mechanism depends on PEO concentration, solvent polarity, and temperature. The coarsening velocity is mainly influenced by the viscosity, which depends on the PEO concentration and molecular weight. The SGT is linked to the hydrolysis and condensation reaction, which depends on pH value and water content [25].

The PEO-induced phase separation does not significantly affect the development of the mesopore system, which is characteristic of alumina sol-gel syntheses and one of its main benefits. As the synthesis procedure itself was only slightly modified compared to the previously published route [26], the reason for the observed diverging pore formation mechanism must lie in the altered composition of the initial reaction solution. No significant change of SGT is observed. Therefore, a change in mechanism and/or coarsening velocity can be related to the change of alcohol-water volume ratio, polarity relating to the methanol/ethanol exchange [33,34,35,36], and/or complexation of PEO with sodium-ions [37,38].

The exchange of ethanol for methanol, the change in the alcohol-water volume ratio, and the addition of sodium-ions seem to have a decisive influence on structure formation and phase separation in the sol-gel process to form sodium-containing alumina monoliths in the first step. To fully elucidate the individual influences, further systematic studies of the individual and combined contributions of these three parameters are required.

### 3.3. Characterization of Xerogels Discs

The employed sol-gel route resulted in defective monoliths. For application in a test cell, the calcined and grounded, polymer-free xerogel was pressed into a disc shape (cf. Figure 8) without further thermal treatment.

The phase composition should, thus, remain unchanged, an additional investigation by XRD was not performed. The samples exhibit a monomodal, open-pore system with a pore size of 16–18 nm (cf. Appendix A), a pore volume of 0.31–0.37 cm^3^∙g^−1^, and a porosity of 52–57%. The pore sizes of the pressed discs are slightly, but distinctly smaller than those of the calcined xerogels. Appendix A summarizes the microstructural data of the pressed discs. The pore system of the starting material likely remained intact, although it may have been compacted, due to the pressurization. Furthermore, the pore volume and porosity are slightly decreased for the same reason.

For comparability, the uncoated discs were calcined according to the procedure described in Section 2.2. to evaluate the changes of phases and microstructure of the discs themselves (cf. Figure 9).

The XRD patterns of the calcined discs show distinct reflexes of α-Al_2_O_3_ and the main reflex of Na-β-alumina at 7.8° 2θ. Na-β’-alumina and Na-β”-alumina are structurally very similar and can be distinguished through the reflexes at 44.5° and 46.0° [39,40]. As the secondary reflexes are very small, no distinction can be made based on the recorded XRD patterns. The relative intensity of the main Na-β-alumina reflex at 7.8° is slightly increased with 0.10 g initial PEO, as confirmed by the subsequent Rietveld analysis. All XRD patterns show a broad diffuse background, indicating that the samples are not completely crystallized. The relative ratio of the crystal phases was determined by Rietveld analysis.

The non-crystalline content was neglected for this calculation. The samples consist mainly of α-Al_2_O_3_ (>90%)_._ The Na-β-alumina content is similar in the samples without and with 0.01 g PEO (2 wt.-% or 1 wt.-%, respectively). The sample with an initial PEO amount of 0.10 g has a higher Na-β-alumina content with 5 wt.-%. Nevertheless, the determined Na-β-alumina amounts are all within the error margins of the method.

The XRD study illustrates that crystallization is not completed. The additional heat treatment at 1400 °C for 2 h promotes the formation α-Al_2_O_3_ from the transition phases. The main difference between the samples is the higher Na-β-alumina content of the sample initially containing 0.10 g of PEO. PEO promotes the formation of alumina rich phase, including a solvent phase, which could be enriched in water and sodium-ions [41]. Furthermore, PEO species could complex sodium-ions [37,38]. Both effects might lead to a better interaction between sodium-ions and the alumina network, such that the conversion to Na-ß-alumina is improved.

The further crystallization leads to a significantly different pore system. Table 1 summarizes the texture data from the mercury intrusion (cf. also Appendix A).

The pore sizes of the calcined discs are increased from meso- to macropores with diameters between 219 and 374 nm, whereas the pore volumes decrease to 0.2–0.3 cm^3^∙g^−1^. The porosity of the PEO free sample decreases (−5%), whereas the porosity of samples initially containing PEO remains constant. The deviation of the pore sizes is clearly beyond the error margins and can be attributed to the disc manufacturing process.

The increase in pore size is due to the formation of α-Al_2_O_3_ via transformation of transition phases. The transformation is accompanied by shrinkage, particle growth, and sintering processes. This causes cracks and consequently larger pores [25,31]. The decrease in pore volume and porosity can be attributed to sintering effects and deviations in the manufacturing process. PEO in the initial reaction solution may prevent pore collapse and improves the porosity of the heat-treated disc. Nevertheless, the mechanism is not yet fully understood and requires further investigation. Furthermore, the results clearly show that improvements in the synthesis procedure and in the conditions of the thermal treatment are necessary to increase the content of Na-β-alumina in the macroporous discs. In comparison, the bilayer electrolytes of Jung et al. [21] contain large parts of ZrO_2_ next to Na-β”-alumina. The porous bulk material contains pores of ≈12 µm, whereas the pore system was not further investigated.

### 3.4. Partially Porous Solid-State Bilayer Electrolyte

Untreated discs with initial PEO contents of 0.00, 0.01, and 0.10 g were used to prepare bilayer electrolytes. The thus obtained partially porous, polymer-free solid-state bilayer electrolytes (BE) were characterized regarding phase composition, microstructure, and ionic conductivity.

The phase composition was evaluated by XRD and subsequent Rietveld refinement.

The number of the Na-β-alumina phases increased, due to coating and subsequent sintering to 9 wt.-% in case of the polymer-free BE and up to 14 wt.-% in case of the BE initially containing 0.10 g PEO. This increase was expected because additional Na-β-alumina was applied via spin coating. Consequently, the α-Al_2_O_3_ amount is decreased by about 10 wt.%. The higher amount of Na-β-alumina in the samples with initial PEO can be attributed to its influence on the phase conversion (cf. Section 3.3).

To further investigate the influence of the coating on the location of Na-β-alumina, the BE were analyzed via SEM imaging. SEM micrographs of the BE synthesized with 0.10 g initial PEO are shown in Figure 10.

In the cross-section of the BE (Figure 10a), three different areas are visible. On the left, there is a dense Na-β-alumina layer with a thickness of about 50 to 100 µm (Figure 10c). The pristine porous bulk material of the disc with a thickness of about 2.2 mm can be seen on the right side of the cross-section (Figure 10a). In between the dense Na-β-alumina layer and the porous disc bulk material, a third intermediate, sodium-enriched layer with a thickness of about 0.3 to 0.4 mm can be identified. Figure 10b shows the transition from the bulk to the intermediate layer.

The thin dense layer (Figure 10d,e) exhibits the typical flake-like shapes of Na-β-alumina crystallites [19,42,43]. Nevertheless, the dense layer shows some unintended porosity, but still enables a sufficiently low leakage rate of less than 1 × 10^−7^ mbar∙L∙s^−1^, while the untreated porous discs had a leakage rate exceeding the detection range of the employed setup. It is not yet fully understood whether the porosity of the dense layer restricts the long-term stability of the bilayer electrolyte.

EDX mapping was performed to clarify the origin of the intermediate layer. Figure 11 displays the Na distribution within the electrolytes.

The dense layer has the highest sodium concentration, while the porous bulk material, as shown in the previous sections, is poor in sodium. The intermediate area is distinguished by a higher sodium concentration originating from sodium diffusion from the dense layer into the porous bulk material. Since the transport of Na-ions in Na-β-alumina is faster than in Al_2_O_3_, Na-enriched domains occur predominantly within the domains already rich in Na-β-alumina, and not homogeneously [44].

A diffusion of other elements contained by the dense layer, like Ti and Zr, could not be observed (cf. Appendix A). In contrast to the other samples, the intermediate area of the sample initially synthesized with 0.10 g PEO exhibits large defects. These are most certainly caused by internal stress during the sintering step, resulting in the phase transition from transition Al_2_O_3_ to porous Na-β-alumina.

At last, the ionic conductivity of the BEs was determined. The results are given in Figure 12.

The ionic conductivity of all samples increases with increasing temperature, due to the enhanced conductivity of Na-β-alumina and NaAlCl_4_. A higher initial PEO amount, and hereby, an increased Na-β-alumina phase content in the disc in combination with the porosity also increased the conductivity. Consequently, the BE with 0.10 g initial PEO has the highest ionic conductivity of 0.10 S∙cm^−1^ at 300 °C, while the sample without PEO only has an ionic conductivity of 0.037 S∙cm^−1^. The stagnation of the conductivity between 280 and 300 °C can possibly be explained by the increasing resistance of sodium. Jung et al. reached a lower conductivity of 0.015 S∙cm^−1^ at 300 °C in a similar measurement setup. The reason for the relatively low conductivity is the high amount of isolating ZrO_2_ (40 vol.%) and a partially closed porosity. [21] Furthermore, the high surface tension of liquid sodium could potentially hinder the infiltration of the porous support.

A study from Zhu et al. [42] showed that the correct number of Na-ions in Na-β-alumina electrolytes is crucial to achieving high ionic conductivity. By reducing the Na_2_O to Al_2_O_3_ ratio from the optimum level of 1/5 to 1/7, the Na-β”-alumina phase content of the synthesized electrolytes dropped, and thereby, the ionic conductivity was reduced from 6.6 × 10^–3^ to 3.7 × 10^–5^ S∙cm^−1^ at 300 °C. This observation can mainly be explained by the collapse of percolation pathways of Na-β-alumina particles and a lack of Na-ions within the Na-β-alumina structure.

Due to the lower Na-β-alumina phase content in the bulk material synthesized with 0.00 and 0.01 g PEO, the conductive paths for Na-ions are limited, while the BE with an initial PEO amount of 0.10 g has more conduction paths. This can be related to the higher Na-β-alumina phase content. As discussed in Section 3.2.1 and Section 3.3, PEO alters the Al_2_O_3_ phase conversion and porosity. The conductivity is, hence, only indirectly influenced by PEO.

Additionally, both the larger amount and size of NalAlCl_4_-filled pores in the samples prepared with higher initial PEO amounts enable a better conductivity within the bulk layer.

However, since the BEs with 0.01 g and 0.10 g initial PEO display similar porosities, but very different ionic conductivities, while the BE with 0.00 g and 0.01 g initial PEO have similar ionic conductivities, but different pore sizes, the role of pores within the three tested samples seems to be minor compared to the phase content. It can be concluded that the ion transport in the bulk layer is not only achieved by the pores filled with NaAlCl_4_, but also by the conductivity of the bulk material itself. High porosity and bulk conductivity are important to achieve a highly conductive electrolyte.

The electrochemical behavior of the material described in this work, motivates further research in the field of bilayer solid electrolytes. Most of the fundamentals, such as the orientation of the porous side within the battery, influence of the pore size and porosity (quantitatively), required thickness of the dense layer, and preparation route, have hardly been explored.

Further, research might focus on increasing the Na-β-alumina phase content through the established method of sodium infiltration or vapor phase conversion [44,45]. The goal is to hereby exceed the conductivity of 0.2 S∙cm^−1^ at 300 °C, which is typical for polycrystalline Na-β-alumina electrolytes.

## 4. Conclusions

In this work, a novel bilayer electrolyte for sodium-ion batteries was prepared and described. By integrating sodium-ions and PEO into the sol-gel synthesis, a mesoporous xerogel is formed, wherein 0.10 g PEO leads to additional spherical cavities by phase separation. Through the further processing of the granulate to a calcined disc, a macroporous α-Al_2_O_3_ disc containing Na-β-alumina could be successfully produced.

The porous disc was successfully spin coated with Na-β-alumina to produce bilayer electrolytes. The electrolytes were characterized regarding phase composition, morphology, and conductivity in an application near (Na/NiCl_2_-cell) environment. The measured conductivity of 0.10 S∙cm^−1^ at 300 °C is plausible compared to other solid electrolytes, but still lower than the typical conductivity of dense bulk Na-β-alumina (about 0.2 S∙cm^−1^ at 300 °C). The results motivate further in-depth studies on how the electrochemical behavior of bilayer electrolytes changes with the preparation method and material properties.

Further research will focus on increasing the Na-β-alumina phase content of the porous bulk material, which was found to be the most promising way to improve the bilayer electrolyte performance.

## Figures and Tables

**Figure 1 materials-14-00854-f001:**
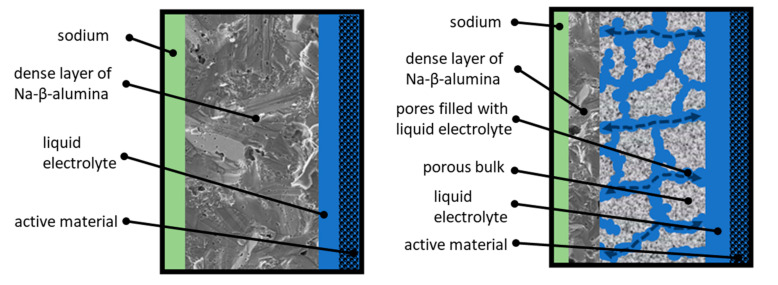
Left: Sketch of the operating principle of a state-of-the-art Na/NiCl_2_-cell. Right: Sketch of the operating principle of a partially porous solid-state bilayer electrolyte.

**Figure 2 materials-14-00854-f002:**
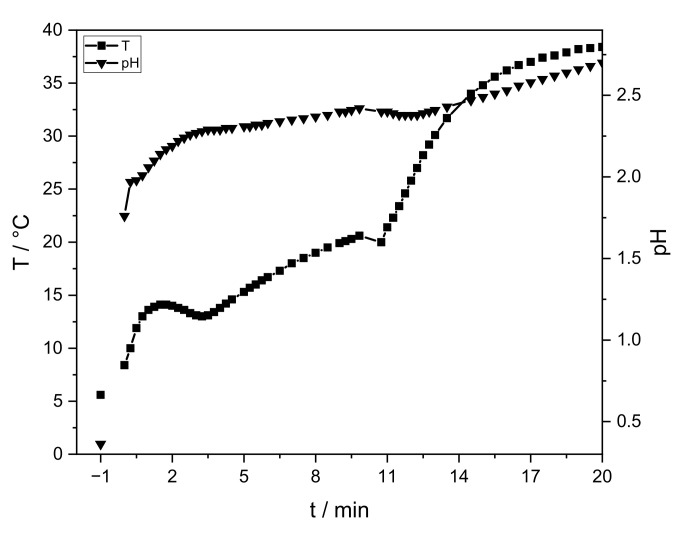
Progression of temperature (squares) and pH (triangles) of the sol-gel process without PEO. The initial temperature and pH value of the AlCl_3_-solution are given at “−1” min. At 0 min sodium hydroxide and PO are integrated. The measurement is stopped after gelation (t = 20 min).

**Figure 3 materials-14-00854-f003:**
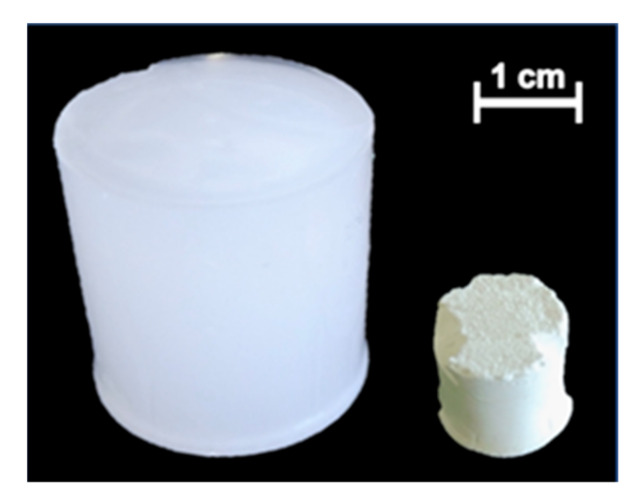
Photograph of the lyogel (left, after aging) and xerogel (right, after drying).

**Figure 4 materials-14-00854-f004:**
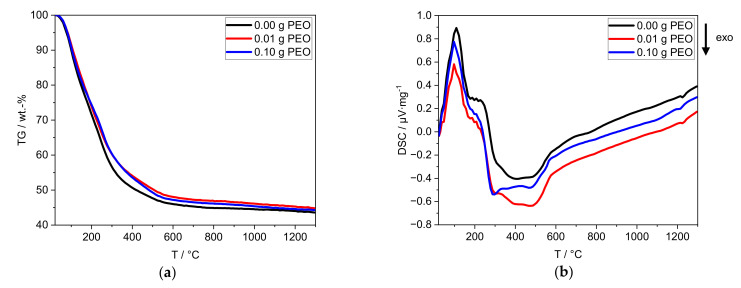
Thermal analyses of the xerogels by (**a**) TG and (**b**) DSC with a heating rate of 10 K∙min^−1^. The sample without PEO is labeled black, the one containing 0.01 g PEO is labeled red, and the one with 0.10 g PEO is labeled blue.

**Figure 5 materials-14-00854-f005:**
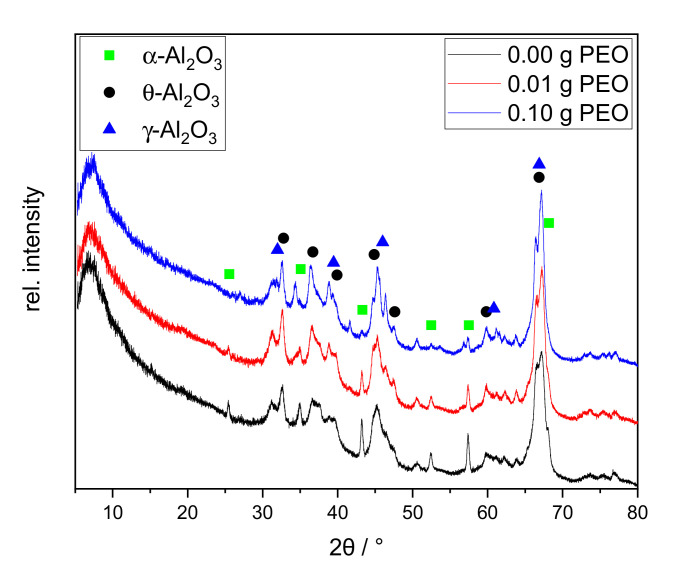
XRD patterns of xerogels calcined at 1200 °C for 1 h with different initial PEO amounts (0.00 g PEO labeled black, 0.01 g PEO red, 0.10 g PEO blue). The main reflexes of alumina phases are labeled as follows: γ-Al_2_O_3_ blue triangle; θ-Al_2_O_3_ black asterisk; α-Al_2_O_3_ green square.

**Figure 6 materials-14-00854-f006:**
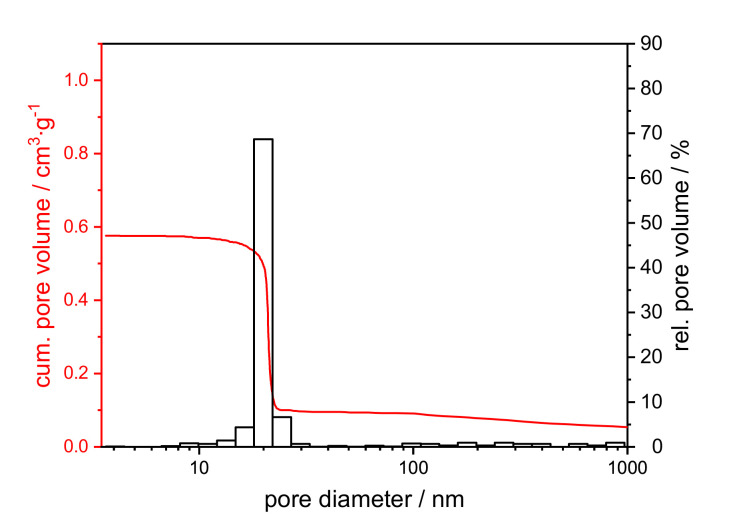
Cumulative (red) and relative pore volume (black) from the mercury intrusion of a xerogel with an initial PEO amount of 0.01 g, calcined at 1200 °C for 1 h.

**Figure 7 materials-14-00854-f007:**
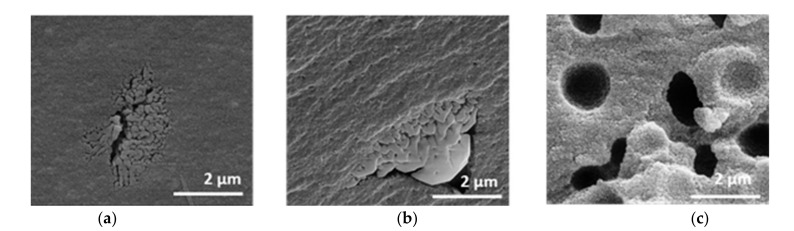
SEM images of xerogels with initial PEO amounts of (**a**) 0.00 g, (**b**) 0.01 g, (**c**) 0.10 g, calcined at 1200 °C for 1 h.

**Figure 8 materials-14-00854-f008:**
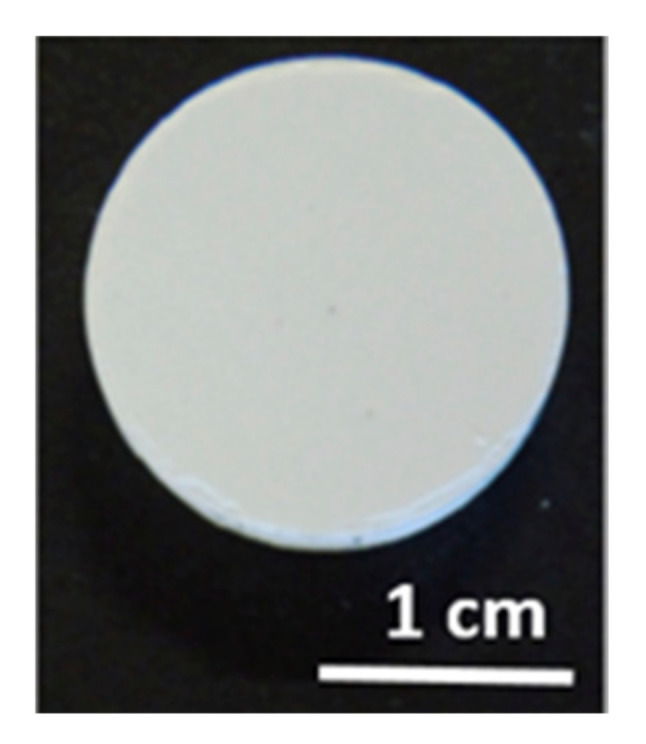
Picture of an untreated disc.

**Figure 9 materials-14-00854-f009:**
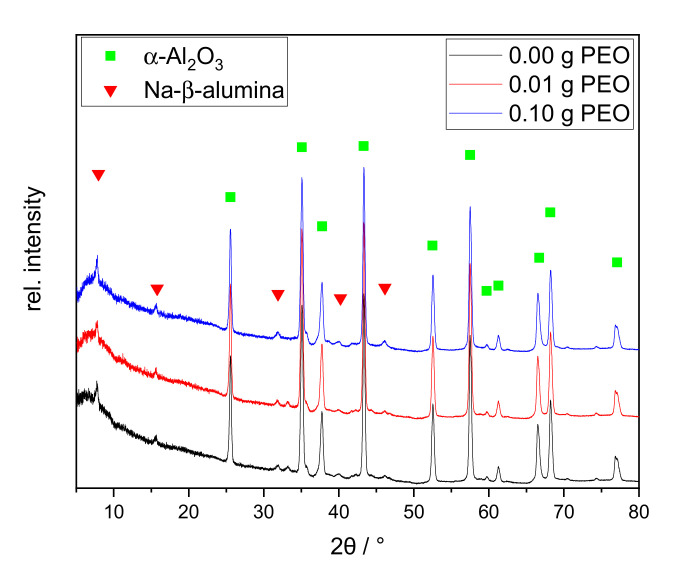
XRD patterns of the discs calcined at 1400 °C for 2 h with different initial PEO amounts: 0.00 g PEO (black), 0.01 g PEO (red), and 0.10 g PEO (blue). The main reflexes of α-Al_2_O_3_ and Na-β-alumina are labeled with a green square and red triangle, respectively.

**Figure 10 materials-14-00854-f010:**
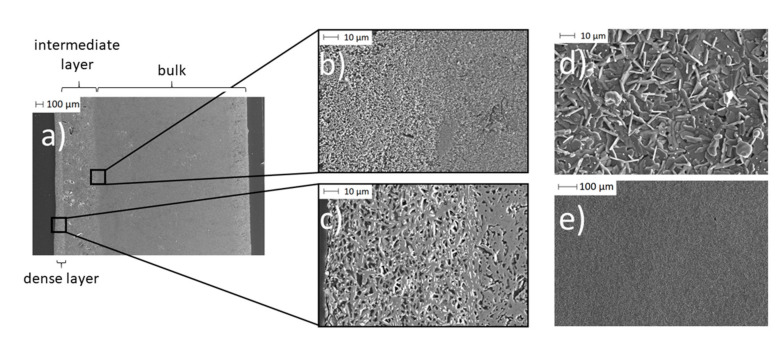
SEM images of a BE synthesized with initial 0.10 g PEO: (**a**) A cross-section of the partially porous solid-state bilayer electrolyte; (**b**) a cross-section of the transition area from the Na-enriched bulk (due to spin coating) to pristine bulk material; (**c**) a cross-section of the dense thin layer; (**d**) and (**e**) surface of the thin layer.

**Figure 11 materials-14-00854-f011:**
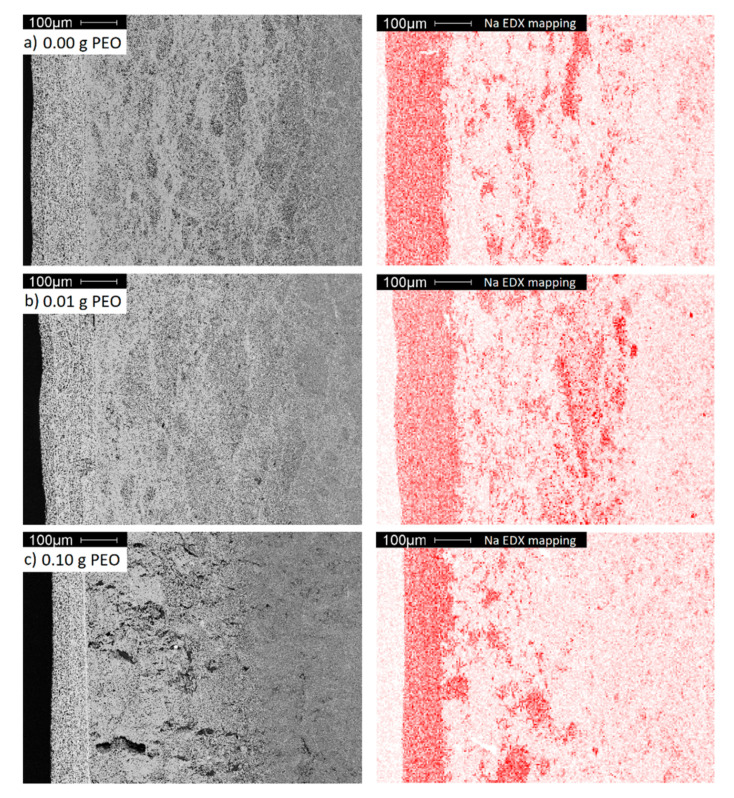
A cross-section SEM images (grey) and EDX mappings of Na (red) of BE synthesized with (**a**) 0.00 g; (**b**) 0.01 g and (**c**) 0.10 g initial PEO.

**Figure 12 materials-14-00854-f012:**
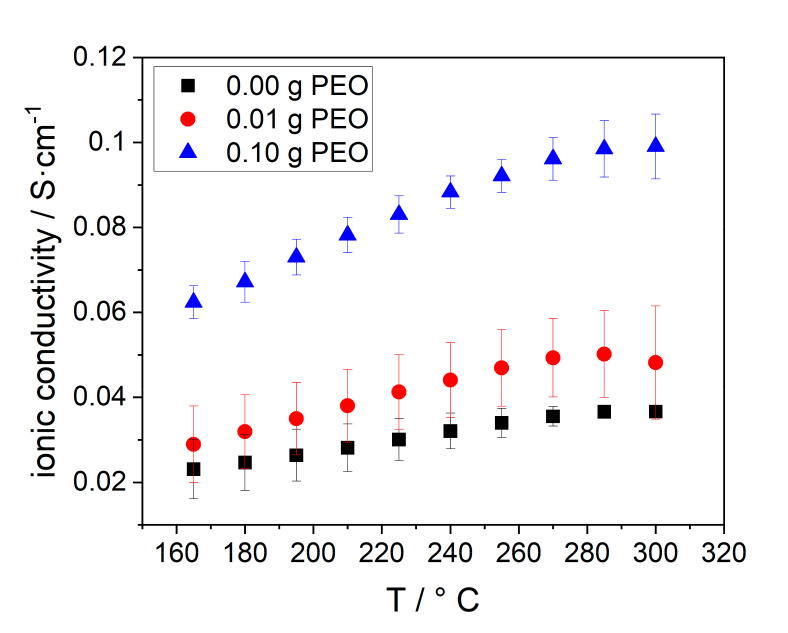
Temperature dependent ohmic conductivity of electrolytes, synthesized with different amounts of initial PEO: 0.00 g (black squares), 0.01 g (red dots), 0.10 g (blue triangles).

**Table 1 materials-14-00854-t001:** The modal pore size (d_mod_), pore volume (V_P_), and porosity (P) determined by mercury intrusion of the heat-treated discs at 1400 °C 2 h with different initial PEO amounts (m_PEO_). Measurement uncertainty. The uncertainty of the measurement is ±5% of the modal pore size, ±0.05 cm^3^∙g^−1^ for pore volume, and ±2% for porosity.

M_PEO_/g	d_mod/_nm	V_P_/cm^3^∙g^−1^	P/%
0.00	257	0.20	47
0.01	374	0.29	55
0.10	219	0.31	57

## Data Availability

The data presented in this study are available on request from the corresponding author.

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
