# Peer review of "Sodium Solid Electrolytes: NaxAlOy Bilayer-System Based on Macroporous Bulk Material and Dense Thin-Film"

_materials, 2021, doi:10.3390/ma14040854_

Round 1

Reviewer 1 Report

The reviewer would like to thank the authors for their submission of the manuscript titled " Sodium solid electrolytes: NaxAlOy bilayer-system based on macroporous bulk material and dense thin-film." The manuscript presents a new idea on bilayer electrolytes for Na metal hydride battery system by replacing solid state Na-β-alumina electrolyte. The focus of the manuscript is on understanding the effect of 3 parameters (methanol addition, Na cations and concentration of PEO) on the crystalline electrolyte formation and the porosity achieved. The manuscript also aims to explore the ionic conductivity for prepared electrolytes.

Based on the manuscript introduction, a clear understanding on how the bilayer system provides a distinct advantage is missing as well as a literature survey on how the proposed system compares to similar systems already in use. Moreover, various statements throughout the text provide no evidence of a clear understanding of the mechanisms involved or a distinct relationship between the explored material properties (porosity and crystalline structure) and relevant performance metrics (ionic conductivity). As such the manuscript cannot be considered for publication in its current form.

The reviewer requests the authors to address both these critical points before a new submission is considered.

Specific comments and concerns include:

  • How does a bilayer system provide an advantage over bulk/polycrystalline Na-β-alumina, since the authors claim that to improve their electrolyte, a larger quantity of Na-β-alumina is needed.
  • What is the advantage of a porous bilayer electrolyte over an electrolyte system of dense Na-β-alumina and liquid NaCl4 without a porous layer.
  • How does the proposed work differ from similar work in literature (Keeyoung Jung, Hee-Jung Chang, Jeffery F. Bonnett, Nathan L. Canfield, Vincent L. Sprenkle, Guosheng Li,
    An advanced Na-NiCl2 battery using bi-layer (dense/micro-porous) β″-alumina solid-state electrolytes, Journal of Power Sources, Volume 396,
    2018, Pages 297-303) 
  • While the differences in porosity is used to explain the effect of PEO, the porosity itself has no relation to realizable properties such as ionic conductivity. This points to the fact that PEO itself may not be a significant contributing factor and other factors maybe needed to present a complete understanding. 
  •  Moreover, the phase separation that is shown to be achieved by PEO and its suppression of α-Al2O3 are not discussed in the context of ionic conductivity and optimization of PEO amount for best bilayer system.

Author Response

Dear Reviewer 1,

we would like to thank you for your detailed feedback to our manuscript “Sodium solid electrolytes: NaxAlOy bilayer-system based on macroporous bulk material and dense thin-film.”

Point 1 How does a bilayer system provide an advantage over bulk/polycrystalline Na-β-alumina, since the authors claim that to improve their electrolyte, a larger quantity of Na-β-alumina is needed.

Response 1: The differences between a porous and dense Na-β-alumina electrolyte are described in lines 51-71 with the potential advantage of bilayer electrolytes regarding the ionic conductivity. Yet, the potential could not be fully used, as the conductivity measurements in chapter 4 showed. The reason for the moderate conductivity was found in the low Na-β-alumina phase content (lines 474-479).

Point 2 What is the advantage of a porous bilayer electrolyte over an electrolyte system of dense Na-β-alumina and liquid NaCl4 without a porous layer

Response 2:  Lines 51-71 describes the potential advantage of bilayer electrolytes regarding the ionic conductivity.

Point 3  How does the proposed work differ from similar work in literature (Keeyoung Jung, Hee-Jung Chang, Jeffery F. Bonnett, Nathan L. Canfield, Vincent L. Sprenkle, Guosheng Li, An advanced Na-NiCl2 battery using bi-layer (dense/micro-porous) β″-alumina solid-state electrolytes, Journal of Power Sources, Volume 396, 2018, Pages 297-303)

Response 3: The advantage of the bilayer system is now more emphasized in the introduction as well as the literature (Jung et al. ) given by you was included (lines 73-86). The following lines 91-109 illustrate the difference with our work. The differences of our pore system to that from Jung et al. is discussed in line 419-421. A comparison of the conductivity to Jung et al. is made in line 473.

Point 4 While the differences in porosity is used to explain the effect of PEO, the porosity itself has no relation to realizable properties such as ionic conductivity. This points to the fact that PEO itself may not be a significant contributing factor and other factors maybe needed to present a complete understanding. 

Point 5 Moreover, the phase separation that is shown to be achieved by PEO and its suppression of 2-Al2O3 are not discussed in the context of ionic conductivity and optimization of PEO amount for best bilayer system.

Response 4 and 5: The influence of PEO on the crystal structure of the bilayer electrolyte is discussed in line 430-432. We further clarified the fact that PEO is removed from the sample by calcination at 1200 °C (lines 292,298, 315, 364, 424). Thus, in line 504-506, we now indicate the indirect influence of PEO on the conductivity.

Kind regards,

Antonia Hoppe

Reviewer 2 Report

The author prepared a novel sodium solid electrolytes based on NaxAlOy bilayer-system, with a good ion conductivity of up to 0.1 S cm-1 at 300 °C. This work provided a new concept of sodium solid electrolytes for next-gen sodium-ion batteries. the overall quality of the paper has reached the publishing standard of Materials, so I would like to recommend to accept after minor revision. some detail suggestion as follow: 1.At line 37, Why is the conduction plane filled with Na+, and the ionic conductivity of these lattice structures is greater than 0.2S cm-1 at 300 degrees? It seems that the reason for this has not been explained clearly, and the benefits of this lattice structure have not been highlighted. 2.The conclusions in lines 90-91 are too suggestive and do not clearly give a clear relationship between characteristics and phase and microstructure. For example, what kind of lattice structure and what size pore structure is conducive to exhibiting this good characteristic. You can add a diagram to better illustrate this point and the connection between them. 3.Figure 5. This picture looks a bit messy, especially those points, which are not uniform and regular. 4.Figure 10. The serial numbers identified above a) b) c) d) are different in size. 5.Figure 12. Why does 0.01gPEO show a downward trend after increasing temperature? 6.In references, some latest important literatures about sodium ion battery and solid electrolyte need to be citied, 1. Sun, Zhen, et al. "Transition metal dichalcogenides in alliance with Ag ameliorate the interfacial connection between Li anode and garnet solid electrolyte." Journal of Power Sources 468 (2020): 228379. 2. Bao, Weizhai, et al. "Boosting Performance of Na–S Batteries Using Sulfur-Doped Ti3C2T x MXene Nanosheets with a Strong Affinity to Sodium Polysulfides." ACS nano 13.10 (2019): 11500-11509. 3. Shanmukaraj, Devaraj, et al. "Highly Efficient, Cost Effective, and Safe Sodiation Agent for High‐Performance Sodium‐Ion Batteries." ChemSusChem 11.18 (2018): 3286-3291.

Author Response

Dear Reviewer 2,

we would like to thank you for your feedback to our manuscript “Sodium solid electrolytes: NaxAlOy bilayer-system based on macroporous bulk material and dense thin-film.”

 Point 1: At line 37, Why is the conduction plane filled with Na+, and the ionic conductivity of these lattice structures is greater than 0.2S cm-1 at 300 degrees? It seems that the reason for this has not been explained clearly, and the benefits of this lattice structure have not been highlighted.

Response 1: At line 37 (now lines 36-38), the explanation of the crystal structure of Na-β-alumina was extended. Further references were added to present a deeper dive into the crystal structure of Na-β”-alumina.

Point 2: The conclusions in lines 90-91 are too suggestive and do not clearly give a clear relationship between characteristics and phase and microstructure. For example, what kind of lattice structure and what size pore structure is conducive to exhibiting this good characteristic. You can add a diagram to better illustrate this point and the connection between them.

Response 2: The conclusion in lines 90-91 (now lines 91) was rephrased. The influence of pore size and phase content is discussed in chapter 4.

Point 3: Figure 5. This picture looks a bit messy, especially those points, which are not uniform and regular.

Response 3: Figures 5 was changed.

Point 4: Figure 10. The serial numbers identified above a) b) c) d) are different in size.

Response 4: Figures 10 was changed.

Point 5:  Why does 0.01gPEO show a downward trend after increasing temperature?

Response 5: The downward trend in figure 12 is due to a the measurements of second batch and averaging this trend is minor now.

Point 6: In references, some latest important literatures about sodium ion battery and solid electrolyte need to be citied

  1. Sun, Zhen, et al. "Transition metal dichalcogenides in alliance with Ag ameliorate the interfacial connection between Li anode and garnet solid electrolyte." Journal of Power Sources 468 (2020): 228379.
  2. Bao, Weizhai, et al. "Boosting Performance of Na–S Batteries Using Sulfur-Doped Ti3C2T x MXene Nanosheets with a Strong Affinity to Sodium Polysulfides." ACS nano 13.10 (2019): 11500-11509.
  3. Shanmukaraj, Devaraj, et al. "Highly Efficient, Cost Effective, and Safe Sodiation Agent for High‐Performance Sodium‐Ion Batteries." ChemSusChem 11.18 (2018): 3286-3291.

Response 6:  The publication of Zhen Sun et al. (1.) was not included in the manuscript because the focus here is on sodium batteries and a comparison to lithium-ion batteries is beyond the scope of our article. The publications from Shanmukarj et al. and Bao et al. (2. And 3.) were cited in the section “Introduction” (lines 40).

Kind regards,

Antonia Hoppe

Reviewer 3 Report

The ms reports an investigation on sodium solid electrolytes and specifically, NaxAlOy bilayer-system based on macroporous bulk material and dense thin-film. Overall, the ms is interesting, well written with a thorough discussion supported by experimental data reported using figures of very god quality. I recommend minor revision as follows:

-Fig. 2: I wonder if the red colour in figure to represent the pH could be avoided and rather using black. also there is no need to report for the pH so many sub-scale values;

-please consider if some of the several tables reported could be moved to the supporting information section;

-How many times the experiment summarised in Fig. 12 has been repeated? perhaps error bars should be introduced.

Author Response

Dear Reviewer 3,

we would like to thank you for your feedback to our manuscript “Sodium solid electrolytes: NaxAlOy bilayer-system based on macroporous bulk material and dense thin-film.”

Point 1: I wonder if the red colour in figure 2 to represent the pH could be avoided and rather using black. also there is no need to report for the pH so many sub-scale values;

Response 1: The color and scaling were adjusted in figure 2 (line 210).

Point 2: Please consider if some of the several tables reported could be moved to the supporting information section;

Response 2: Tables 1, 2, and 3 have been moved to the supplementary information and the references were adjusted (lines 136, 306, 377). The tables 4 and 6 have been deleted.  Consequently, the table headers of tables 1, S2 and S3 were corrected.

Point 3: How many times the experiment summarised in Fig. 12 has been repeated? perhaps error bars should be introduced.

Response 3: While the first review, the measurements of the ionic conductivity (Figure 12) were only carried out on one sample. For the current version, the measurements were repeated on a second batch. Error bars are now introduced.

Kind regards,

Antonia Hoppe

Round 2

Reviewer 1 Report

The reviewer would like to thank the authors for the revision of the manuscript titled " Sodium solid electrolytes: NaxAlOy bilayer-system based on macroporous bulk material and dense thin-film" and the response to the reviewer comments.  A better understanding on the advantage of bilayer systems and earlier concerns with regards to similar work in literature have been addressed with scope for further improvement.

 After careful review, a few additional details still need to be determined and major and minor revisions are suggested as follows:

  1. A clear goal of the manuscript (as stated in the introduction) is to improve on existing all solid state Na-β- alumina electrolytes by improving the conductivity for the same thickness. This motivation has been added to the manuscript, as well as the information provided that this goal was not achieved. However, the role of Na-β- alumina , present in the porous network, could also affect the conductivity for the same liquid electrolyte used. This is clearly demonstrated through the results in the manuscript but is missing as an important motivation factor on why new methods need to be developed to improve the bilayer system. The reviewer would recommend the authors to consider this addition as a strong differentiating point from previous work. This is mentioned on Page 15 Line 496-498 but would greatly benefit being introduced earlier on.
  2. Related to the first point and new text added, while a comparison with Jung et al, is provided in the revision, any advantages in terms of cost, synthesis time or otherwise for the chosen route would be beneficial to include.
  3. Page 3 Line 105 and 106 mentions that the other goal of the manuscript is to investigate the effect of PEO concentration, sodium cations and methanol on the synthesis. However, apart from PEO concentration no conclusive evidence is provided for the role sodium cations directly and for methanol on the synthesis. The sodium cation effect is indirect from the different Na-β- alumina as a result of PEO weight and can be considered a PEO concentration effect. The EDX in Figure 11 for Na also does not provide any conclusive Na cation trends. The reviewer would like to request the authors to consider removing methanol and sodium cation effects from the introduction and instead focus on PEO concentration.
  4. Page 7 Line 243-244, a systematic shift with higher PEO content cannot be clearly seen from Figure 4b. Could the authors include an inset of Figure 4b in the main text or supplementary to support this claim?
  5. Page 9 Line 336-337, could the authors comment on how PEO induced pore volume when within uncertainty of measurement can lead to significant changes in the pore structure as seen from Figure 7. Could the measurement be erroneous, or possibly not sufficient images on SEM (from different areas on the same sample) to make bulk conclusions?
  6. Page 11 Line 384-385, the trend for relative intensity of the main Na- ?-alumina reflex at 7.8 increasing with higher PEO amount is not clearly visible from Figure 9. Since this is one of the crucial evidences for the effect of PEO, an inset of the peak around 7.8 should be included in Figure 9 to support the claim and the story.
  7. Since the results on the effect of porosity on Page 14 Line 493-496 seem counterintuitive, the reviewer would like to ask about the statistics of the performed experiments. How many films were measured for the conductivity? And how many batches of the discs were prepared for this study? Are there expected variations in the conductivity for the 3 PEO concentrations that could affect the interpretation of the conductivity results?
  8. As a minor curiosity, for the mechanism involving PEO, as the authors mention a control on the exothermic process, it seems likely that the boehmite formation/agglomeration to larger sizes is a thermal effect. PEO being an organic system, could lead to poor thermal conductivity in the system and a thermal suppression could potentially be achieved by a thermally absorbing salt such as NaCl also? Could the authors comment on this?

Author Response

Dear Reviewer 1,

we would like to thank you for your feedback on our manuscript “Sodium solid electrolytes: NaxAlOy bilayer-system based on macroporous bulk material and dense thin-film.”
Please find below a detailed response to your remarks.

Point 1: A clear goal of the manuscript (as stated in the introduction) is to improve on existing all solid state Na-β- alumina electrolytes by improving the conductivity for the same thickness. This motivation has been added to the manuscript, as well as the information provided that this goal was not achieved. However, the role of Na-β- alumina, present in the porous network, could also affect the conductivity for the same liquid electrolyte used. This is clearly demonstrated through the results in the manuscript but is missing as an important motivation factor on why new methods need to be developed to improve the bilayer system. The reviewer would recommend the authors to consider this addition as a strong differentiating point from previous work. This is mentioned on Page 15 Line 496-498 but would greatly benefit being introduced earlier on.

Response 1: This is now also mentioned in the abstract (lines 21-24) and in the introduction (line 67-68). The motivation factor is also mentioned in the lines 91-94.

Point 2: Related to the first point and new text added, while a comparison with Jung et al, is provided in the revision, any advantages in terms of cost, synthesis time or otherwise for the chosen route would be beneficial to include.

Response 2:  In line 91-94 a short comparison to Jung is given. Since we are still at an early point regarding the research of bilayer electrolytes, there are no financial calculations available for the process discussed in this work, nor for the process of Jung et al.

Point 3: Page 3 Line 105 and 106 mentions that the other goal of the manuscript is to investigate the effect of PEO concentration, sodium cations and methanol on the synthesis. However, apart from PEO concentration no conclusive evidence is provided for the role sodium cations directly and for methanol on the synthesis. The sodium cation effect is indirect from the different Na-β- alumina as a result of PEO weight and can be considered a PEO concentration effect. The EDX in Figure 11 for Na also does not provide any conclusive Na cation trends. The reviewer would like to request the authors to consider removing methanol and sodium cation effects from the introduction and instead focus on PEO concentration.

Response 3: In line 16 methanol and sodium cations were deleted.

Point 4: Page 7 Line 243-244, a systematic shift with higher PEO content cannot be clearly seen from Figure 4b. Could the authors include an inset of Figure 4b in the main text or supplementary to support this claim?

Response 4: The exothermic process from Fig. 4b is shown enlarged in a new Fig. S1. The numbering of the supplementary information has been adjusted.

Point 5: Page 9 Line 336-337, could the authors comment on how PEO induced pore volume when within uncertainty of measurement can lead to significant changes in the pore structure as seen from Figure 7. Could the measurement be erroneous, or possibly not sufficient images on SEM (from different areas on the same sample) to make bulk conclusions?

Response 5:  The sentence “PEO induced additional pore volume” in line 342 was deleted. The error due to the measurement method is explained in detail in lines 341-345. Enough SEM images were taken and from different charges, so that a statistical error can be excluded.

Point 6: Page 11 Line 384-385, the trend for relative intensity of the main Na- ?-alumina reflex at 7.8 increasing with higher PEO amount is not clearly visible from Figure 9. Since this is one of the crucial evidences for the effect of PEO, an inset of the peak around 7.8 should be included in Figure 9 to support the claim and the story.

Response 6 In line 395, a reference was given to the subsequent Rietveld analysis.

Point 7: Since the results on the effect of porosity on Page 14 Line 493-496 seem counterintuitive, the reviewer would like to ask about the statistics of the performed experiments. How many films were measured for the conductivity? And how many batches of the discs were prepared for this study? Are there expected variations in the conductivity for the 3 PEO concentrations that could affect the interpretation of the conductivity results?

Response 7: The conductivity was measured on two samples from two different batches. A minor variation is expected, because different batches were used, but the results are reproducible within the limits of the method.

Point 8: As a minor curiosity, for the mechanism involving PEO, as the authors mention a control on the exothermic process, it seems likely that the boehmite formation/agglomeration to larger sizes is a thermal effect. PEO being an organic system, could lead to poor thermal conductivity in the system and a thermal suppression could potentially be achieved by a thermally absorbing salt such as NaCl also? Could the authors comment on this?

Response 8: The samples no longer contain carbon after thermal treatment. Thus, a direct influence on the conductivity can be excluded. CHN analysis (lines 169-171) was added in lines 267-269 to demonstrate that the remaining carbon content is negligible (0.22 wt.-%). Evidently, the possibility of achieving products resembling the ones reported here via a different route, specifically with different additives, is conceivable. However, discussion thereof would go far beyond the scope of our article manuscript.

Kind regards,

Antonia Hoppe